# g-C_3_N_4_ Based Photocatalyst for the Efficient Photodegradation of Toxic Methyl Orange Dye: Recent Modifications and Future Perspectives

**DOI:** 10.3390/molecules28073199

**Published:** 2023-04-04

**Authors:** Abdulelah Aljuaid, Mazen Almehmadi, Ahad Amer Alsaiari, Mamdouh Allahyani, Osama Abdulaziz, Abdulaziz Alsharif, Jawaher Amer Alsaiari, Magdi Saih, Rema Turki Alotaibi, Idrees Khan

**Affiliations:** 1Department of Clinical Laboratory Sciences, College of Applied Medical Sciences, Taif University, P.O. Box 11099, Taif 21944, Saudi Arabia; Ab.aljuaid@tu.edu.sa (A.A.); ahadamer@tu.edu.sa (A.A.A.); m.allahyani@tu.edu.sa (M.A.); o.osama@tu.edu.sa (O.A.); sharif@tu.edu.sa (A.A.); jawaher.a@tu.edu.sa (J.A.A.); magdi-206@hotmail.com (M.S.); rema-rema@hotmail.com (R.T.A.); 2School of Life Sciences, University of Nottingham, Queen’s Medical Centre, Nottingham NG7 2UH, UK; 3School of Chemistry and Chemical Engineering, Northwestern Polytechnical University, Xi’an 710072, China

**Keywords:** photodegradation, photocatalyst, g-C_3_N_4_, modifications, methyl orange

## Abstract

Industrial effluents containing dyes are the dominant pollutants, making the drinking water unfit. Among the dyes, methylene orange (MO) dye is mutagenic, carcinogenic and toxic to aquatic organisms. Therefore, its removal from water bodies through effective and economical approach is gaining increased attention in the last decades. Photocatalytic degradation has the ability to convert economically complex dye molecules into non-toxic and smaller species via redox reactions, by using photocatalysts. g-C_3_N_4_ is a metal-free n-type semiconductor, typical nonmetallic and non-toxici polymeric photocatalyst. It widely used in photocatalytic materials, due to its easy and simple synthesis, fascinating electronic band structure, high stability and abundant availability. As a photocatalyst, its major drawbacks are its limited efficiency in separating photo-excited electron–hole pairs, high separated charge recombination, low specific surface area, and low absorption coefficient. In this review, we report the recent modification strategies adopted for g-C_3_N_4_ for the efficient photodegradation of MO dye. The different modification approaches, such as nanocomposites and heterojunctions, as well as doping and defect introductions, are briefly discussed. The mechanism of the photodegradation of MO dye by g-C_3_N_4_ and future perspectives are discussed. This review paper will predict strategies for the fabrication of an efficient g-C_3_N_4_-based photocatalyst for the photodegradation of MO dye.

## 1. Introduction

Polluted wastewater containing industrial discharge is one of the main causes of the irreversible degradation of ecosystems [1]. Water pollution is considered a serious concern to the global community, since wastewaters from the textile, leather, food and chemical industries discharge hazardous dyes [2]. Organic dyes are the common coloring agents in textile, cosmetics, leather, paper, plastic, printing, rubber and pharmaceutical industries [3], and their use results in severe water pollution and create environmental and esthetic issues [4]. Most of these dyes are toxic, teratogenic, carcinogenic, xenobiotic, and non-biodegradable, owing to their complex structures and large size, and their accumulation create potential threats and risks to human and aquatic life [5,6]. Their continuous discharge in wastewaters into the natural aquatic environment causes non-aesthetic pollution and eutrophication, and can generate very toxic byproducts via oxidation, hydrolysis, or other chemical reactions occurring during water treatment [7]. Dyes affect the central nervous system, liver, kidney, skin, enzymatic system, chromosomes and reproductive systems of human [8]. Dye concentrations in textile wastewater are reported over a wide range of values. A study indicated that the level of dye in the textile effluent is 10–50 mg/L. The concentrations of reactive dyes in cotton factories are reported as 60 mg/L, as well as between 100 and 200 mg/L [9]. Cationic dyes are very toxic even at trace levels, e.g., the concentration of malachite green should not exceed 1.0 μg·L^−1^ in drinking water and 100 μg·L^−1^ in potable waters [10]. Similarly, the permissible levels of permitted food coloring Sunset yellow FCF dye are 100–200 mg/kg [11].

Methyl orange (MO) is an anionic azo dye, having high chemical stability [12], and it belongs to the sulphonated azo group [13]. The IUPAC name of MO is Sodium 4-[(4-dimethylamino) phenyldiazenyl] benzenesulfonate, having a molecular formula of C_14_H_14_N_3_NaO_3_S, a molar mass of 327.33 g/mole, and a density of 1.28 g/cm^3^ [14]. MO is partially soluble in normal water, highly soluble in hot water, and insoluble in ethanol. It is an orange–yellow powder and its melting point is greater than 300 °C [15]. MO can function as week acid, showing approximately 6.5 pH when dissolved in water, and it also works as a pH indicator (color changes from red to yellow in the range of 3.1–4.4) [16]. MO displays a red color in the acidic medium, and in basic media it shows an orange color [17]. The azo group, sulfur group and aromaticity of MO give it an intense coloration, high chemical stability, toxicity and low biodegradability [18]. The chemical structure and optimized structure of MO dye is shown in the Figure 1 [19]. MO dye has various industrial and laboratory applications [20].

MO is extensively used in dyeing, leather, textile, pulp, paper and printing industries, and in research laboratories [21,22]. MO is used widely in the textile industry and is the major constituent of industrial waste discharge, polluting various water bodies [23]. The concentration of MO dye can reach up to 500 ppm in the textile effluent [24]. MO dye is mutagenic, carcinogenic and toxic to aquatic organisms [25,26]. MO is one of the most extensively used hazardous anionic azo dyes, is a highly recalcitrant and refractory xenobiotic, and causes a significant burden in the ecosystem [27]. Azo dyes show resistance in the natural environment, and their bio-degradation has serious hurdles [28]. MO is a more harmful dye, and cause serious water pollution when released into the environment. Its presence at maximum amounts in drinking water produces anemia, abdominal pain, headache, dizziness, mental confusion, excessive sweating and nausea [23]. Acute exposure to MO dye can cause shock, vomiting, cyanosis, increased heart rate, quadriplegia, tissue necrosis and jaundice in humans [29]. MO accidentally enters the body via ingestion, where intestinal microorganisms metabolize it into aromatic amines, which can cause intestinal cancer [30]. Thus, MO-containing wastewater should be decolorized and detoxified before being discharged into the environment [31].

Methyl orange is difficult to be decomposed using conventional methods at ambient conditions, as these methods can accumulate pollutants instead of decomposing them [32]. MO dye is nonbiodegradable, due to its complex aromatic structure and xenobiotic properties [33]. Several methods are described for MO removal, such as adsorption [34,35,36], biodegradation [37], phytoremediation [38] ozonation [39], coagulation/flocculation [40] and heterogenous photodegradation [41,42,43,44]. It has complex molecular structures, which make their removal from wastewater difficult when using conventional treatment methods [45]. Heterogeneous photodegradation has advantages over the other reported conventional techniques due to its process simplicity, complete pollutant mineralization, cost-effectiveness, no/fewer harmful byproduct production and its ability to be carried out at ambient temperature and pressure [46]. The heterogeneous photodegradation (PD) process usually requires a photocatalyst or semiconductor, light source, reactor system and the pollutant and oxygen [47].

Various photocatalysts are used for the PD of MO dye e.g., ZnO-rGO nanorods [48], CuO nanoparticles [29], Au/TiO_2_ nanoparticles [49], Ag–Ni and Al–Ni nanoparticles [50], silver nanoparticles/amidoxime-modified polyacrylonitrile nanofibers [51], g-C_3_N_4_ and B-doped g-C_3_N_4_ [52], SGCN/Fe_3_O_4_/PVIs/Pd [53], etc.

g-C_3_N_4_ is a metal free n-type semiconductor polymer, having several exceptional properties, such as unique structural, optical, electric and physiochemical properties, making g-C_3_N_4_-based materials an emerging class of materials that can be used in various photocatalytic applications [54]. The g-C_3_N_4_ can respond to the visible light (up to 460 nm), due to its suitable band gap of about 2.7 eV. Its conduction band potential (ECB) is about −1.1 V (vs. NHE), and, thus, its photogenerated electrons have a very strong reduction ability [55]. g-C_3_N_4_ is a typical, nonmetallic and non-toxic polymeric photocatalyst, which has attracted tremendous attention for environmental protection because of its easy synthesis, low cost, excellent photochemical stability, favorable band positions [56,57,58], etc. The use of g-C_3_N_4_, from an economical point of view, is considered as a viable choice in the photocatalysis field, because of its intrinsic properties and favorable internal qualities, such as its electrical conductive, photonic, distinctive 2D-stacked hierarchical features, high physical and chemical durability and non-toxicity [59]. Figure 2 illustrates the triazine (C_3_N_3_) and tri-striazine/heptazine (C_6_N_7_) rings, which are the basic structural tectonic units of g-C_3_N_4_ [60]. g-C_3_N_4_ is a layered structure, having a large specific surface area, electron-rich surface, and suitable band gap energy of 2.7 eV, which promotes its photocatalytic efficiency [61]. g-C_3_N_4_ composite-based materials are utilized widely in photocatalytic applications, owing to the low prices of their raw material, easy and simple synthesis, fascinating electronic band structure, high stability, as well as abundant availability [62,63]. g-C_3_N_4_ was synthesized, utilizing inexpensive nitrogen-rich substances (cyanamide, melamine, dicyandiamide, urea and thiourea) as precursors, via self-condensation by deamination, through a thermal reaction [64].

Several reviews are reported in terms of the various aspects of g-C_3_N_4_-based photo-catalysts, such as their design strategies, properties, photocatalytic applications [65], and artificial photosynthesis, as well as their environmental remediation [66], modification with MXenes, the use of their derivatives for various photocatalytic applications [67], recent advances for photocatalytic CO_2_ reduction [68], their use in the disinfection of water and microbial control [69], quantum dot-modified g-C_3_N_4_-based photo-catalysts for different photo-catalytic applications [70], g-C_3_N_4_–metal oxide based nano-composites for photo-catalysis and sensing applications [60], the design and application of active sites in g-C_3_N_4_-based photo-catalysts [71], synthesis and structural developments in visible-light-active g-C_3_N_4_-based photo-catalysts [72], nonmetal modulation of morphology and composition of g-C_3_N_4_-based photo-catalysts [73], etc. There are no clear reviews reported specifically on recent modifications of g-C_3_N_4_ for dye photodegradation. However, a lot of work has been done on dye degradation using g-C_3_N_4_, hence, the present review is further specified to the recent modification in g-C_3_N_4_ for the efficient photodegradation of MO dye. This review will help those researchers who want to study the photodegradation of MO, by applying g-C_3_N_4_. The Scopus database indicates that limited literature is available on the photodegradation of MO dye using g-C_3_N_4_. Figure 3 shows the number of articles published per year on the photodegradation of MO dye by g-C_3_N_4_.

In photodegradation, major drawbacks of g-C_3_N_4_ molecules are the limited efficiency of separating photo-excited electron–hole pairs, a narrow spectral absorption range, high-charge carrier recombination, insufficient sunlight absorption, low specific surface area, as well as a low absorption coefficient [74,75,76,77]. Several modifications have been attempted for improving the photo-catalytic activity of pure g-C_3_N_4_ for the efficient photodegradation of MO dye. Some of them are summarized below.

## 2. Composites and Heterojunctions

g-C_3_N_4_ composite materials were prepared with different materials for enhancing their photodegradation efficiency. Heterojunction systems were developed for improving the photo-catalytic activity of g-C_3_N_4_, enhancing its oxidation and reduction capabilities by utilizing the negative conduction band (CB) of one component, with the more positive valence band (VB) of the other component [78]. The ZnO/g-C_3_N_4_ composite (ZnO/g-C_3_N_4_-20 wt%) exhibited the highest photo-catalytic activity and good recyclability when compared to a pure ZnO catalyst and g-C_3_N_4_ catalyst [62]. The 5 wt% CuO/g-C_3_N_4_ nanolayer composites exhibited very efficient activity and removed 99.7% of the MO dye in 4 min, owing to their nano-size and porous nature [79]. MoS_2_/TiO_2_ heterostructure were decorated on g-C_3_N_4_ nanosheets to obtain a g-C_3_N_4_@MoS_2_/TiO_2_ nanocomposite photocatalyst, which shows a highly efficient removal of (98%) of MO dye in visible light and long-term stability when compared to neat g-C_3_N_4_ and MoS_2_/TiO_2_. The improved photo-catalytic activity of the g-C_3_N_4_@MoS_2_/TiO_2_ nanocomposite could be attributed to its strong response to visible light, the effective separation of photogenerated electron-hole pairs and the narrowed band gap. The catalyst caused a reduction of the C/C0 value of just 2.5%, from the first to the fifth cycle experiment [80]. Similarly, the AgBr/g-C_3_N_4_ composite system represents one with enhanced photo-catalytic activity when compared to mono-component systems under visible light irradiation. In this study, the composite AgBr:g-C_3_N_4_, having a mole ratio of 2:1, displayed efficient activity, and this ratio confirmed that the recombination of e^−^/h^+^ is the rate-limiting step in the coupled system. The re-used catalyst demonstrated good activities by being used in four successive runs without any significant loss in its activity [81]. A ternary ZnO/Fe_3_O_4_/g-C_3_N_4_ composite (magnetic recyclable) efficiently degraded MO dye. Among these composites, the ZnO/Fe_3_O_4_/g-C_3_N_4_-50% composite was found to be very effective, and degraded MO dye more effectively than pure ZnO and pure g-C_3_N_4_. There were three reasons discussed in terms of their enhanced photodegradation efficiency. Their UV-Vis spectra confirmed the strongest visible light response intensity of this composite. The electrochemical and PL results revealed that the rate of recombination of e^−^/h^+^ pair was the lowest. The electrochemical results showed that the photoelectron transfer rate was fastest in this composite than that observed in single components. The ZnO/Fe_3_O_4_/g-C_3_N_4_-50% composite exhibited a higher photocatalytic activity after five recycles [82]. Some g-C_3_N_4_ composites reported for the efficient PD of MO dye are AgBr:g-C_3_N_4_ [83], g-C_3_N_4_/TNTs heterojunctions [84], ternary g-C_3_N_4_/ZnO–W/M nanocomposites [85], g-C_3_N_4_-TiO_2_ nanocomposite [86], CdS/g-C_3_N_4_ hybrid nano-photocatalysts [87], MoO_3_–g-C_3_N_4_ composites [88], g-C_3_N_4_/ZnO composites [89], CoFe_2_O_4_/g-C_3_N_4_ nanocomposites [90], g-C_3_N_4_/Bi_4_O_5_I_2_ [91], ternary g-C_3_N_4_/Ag/γ-FeOOH photocatalysts [92], etc. Some g-C_3_N_4_-based composites/nanocomposites used for the efficient PD of MO dye are consolidated in Table 1.

Apart from these, some metals/metallic compounds, such as Au [105], CoO_x_ [106], Co_3_O_4_ [107], ZnO [108], SrTiO_3_ [109], Gd_2_O_3_ [110], and MnOx [111], were coupled to form g-C_3_N_4_ composite/heterojunctions for enhanced photodegradation of MO dyes.

Similarly, g-C_3_N_4_ quantum dot-based nanocomposites are also reported for the efficient photodegradation of MO dye. The g-C_3_N_4_ QDs/BiPO_4_ nanocrystal composite efficiently degraded about 92%, while g-C_3_N_4_ alone only degraded 75% in 180 min [112]. In another study, MoS_2_/Fe_3_O_4_/g-C_3_N_4_QDs nanocomposite degraded about 99.68% MO dye in 60 min under visible light [113]. Other g-C_3_N_4_ quantum dot-based nanocomposites reported for MO degradation are CaAl_2_O_4_:Eu^2+^,Nd^3+^ phosphor-coupled g-C_3_N_4_ quantum dot composites [114].

The photocatalytic enhancement, achieved by coupling g-C_3_N_4_ with other materials, can be easily understand from the mechanism of photodegradation of MO by BiOI/AgI/g-C_3_N_4_ composites. The photocatalyst exhibited a sandwich-type structure, with AgI sandwiched between g-C_3_N_4_ and BiOI. The CB levels of AgI, BiOI and g-C_3_N_4_ are about − 0.55 eV, 0.58 eV and − 1.12 eV (vs NHE), respectively, and the VB levels of AgI, BiOI and g-C_3_N_4_ are about 2.25 eV, 2.31 eV and 1.57 eV, respectively. Irradiating under visible light, AgI, BiOI and g-C_3_N_4_ become excited and generate e^−^ and h^+^. The photogenerated e^−^ present in the CB of g-C_3_N_4_ will freely transfer to the CB of AgI, and then transfer into the CB of BiOI. Similarly, h^+^ present in the VB of BiOI migrates to the VB of g-C_3_N_4_, using the pathway of AgI. Some photo-induced e^−^ present in CB of g-C_3_N_4_ directly transfers to the CB of BiOI via contacted interfaces, however, a higher barrier height between the g-C_3_N_4_ and BiOI would hinder the process partially. Thus, the stepwise transfer of charge occurres, which results in the efficient transfer of e^−^ and the enhanced separation of created charges. The e^−^ reacts with O_2_ and generates ^•^O_2_^−^ that degrades MO dye efficiently, while the h^+^ can oxidize, directly, the MO dye molecules through the hole oxidation pathway. The VB edge potentials of BiOI, AgI, and g-C_3_N_4_ in the BiOI/AgI/g-C_3_N_4_ composites are insufficient for the formation of ^•^OH radicals. The proposed possible MO degradation reaction mechanisms are represented schematically in the Figure 4 [115].

## 3. Doping of g-C_3_N_4_

As g-C_3_N_4_ demonstrated poor catalytic efficiency due to low light adsorption and high recombination of the separated charges, which can be overcome by doping, in order to increase their conductivity. The LUMO and HOMO of the g-C_3_N_4_ are controllable. Owing to their tunable bandgap, the g-C_3_N_4_ photocatalytic efficiency is affected. Therefore, g-C_3_N_4_ can be modified by doping with elements. Doping is the intentional inserting of impurities into the certain semiconductor in order to tune its bandgap [116]. Sensitized doping can improve the catalytic activity of g-C_3_N_4_ photocatalysts [117]. For example, sulfur fluoride-doped g-C_3_N_4_ (F-SCN) shows enhanced MO degradation when compared to sulfur-doped g-C_3_N_4_ (SCN) and bulk g-C_3_N_4_ (BCN), as concluded from the Figure 5. Figure 5a shows that the F-SCN fluorescence intensity is the lowest, indicating a reduction in the created charges’ recombination rate to a certain extent, an increase in their numbers, and an improvement in its photocatalytic performance. The impedance spectrum (EIS) in the Figure 5b revealed that the radius of F-SCN is smaller than that of SCN and BCN, which revealed that F-SCN demonstrated the highest rate of electron transport. Similarly in the transient photocurrent response study, as represented in the Figure 5c, F-SCN displayed an enhanced photocurrent response, which suggests that the e^−^h^+^ photoexcitation in F-SCN materials requires a longer time, and their utilization time can also be extended as compared to SCN and BCN [118].

Doping of g-C_3_N_4_ with W improved its photocatalytic efficiency and degraded 99.6% within 60 min, using visible light [119]. Introducing boron into the structure of g-C_3_N_4_ will narrow its band gap, in order to absorb more visible light [52]. Boron-doped rGO/g-C_3_N_4_ nanocomposites (B-5%rGO/g-C_3_N_4_) approximately degraded 100% of the MO dyes within 240 min [120]. The P and S-codoped g-C_3_N_4_ enhanced the light absorption capability, surface area, and the charge separation efficiency, and showed higher catalytic activity. The pure g-C_3_N_4_, P-doped g-C_3_N_4_ and S-doped g-C_3_N_4_ degraded 15.26%, 22.67% and 24.86%, while the P and S codoped g-C_3_N_4_ (PSCN-50) degraded 73.25% of MO dye in 60 min under visible light irradiation. The PSCN-50 sample had no obvious activity even after five cycling runs, indicating high stability. [121]. BiVO_4_/pyridine-doped g-C_3_N_4_ shows enhanced performance when compared to the individual components, and photodegraded 97% of MO molecules within 150 min under visible light irradiation. The BiVO_4_/pyridine-doped g-C_3_N_4_ photocatalyst did not show an obvious reduction of activity, even after five cycling runs, implying the stability of photocatalysts [122]. A g-C_3_N_4_ molecule was reconstructed as a g-C_3_N_4_ nanoring by adding natural pollen, which results in abundant heteroatom (C)-doping, increase surface area and the formation of a porous hollow structure. The C-doped g-C_3_N_4_ demonstrated 2.8 times higher MO degradation activity than that of bulk g-C_3_N_4_, and the sample H–CN200–C degraded 95.5% dye after 120 min [123].

In some studies, both doping and coupling are applied to form doped g-C_3_N_4_ heterojunctions, in order to enhance MO dye degradation. For example, g-C_3_N_4_ was doped with Cl and then coupled with Bi_2_WO_6_ to obtain ClCN/Bi_2_WO_6_ heterojunctions. In this study, Bi_2_WO_6_ and ClCN degraded only 20% and 30%, while 10% ClCN/Bi_2_WO_6_ degraded 99% MO dye in 40 min. The enhanced photocatalytic efficiency was credited to the creation of S-scheme heterojunctions, which inhibited the separated charges’ recombination, but accelerated the recombination of the relatively useless holes and electrons [124]. Other such g-C_3_N_4_-based photocatalysts for MO dye degradation reported are B-doped g-C_3_N_4_/MoO_3_ [125], Fe-doped ZrO_2_ nanoparticle-supported g-C_3_N_4_ hybrids [126], CeO_2_/P-C_3_N_4_ composites [127], etc.

## 4. Crystal Phase Control and Defects Introduction

Modification of the bandgap for improved photocatalytic efficiency can also be achieved by variations in the crystal phase, as well as facet control [78]. The crystallinity of g-C_3_N_4_ has been altered by incorporating Ag with g-C_3_N_4_ via calcination for 8 h. The obtained Ag-decorated g-C_3_N_4_ achieved the degradation of MO, and degraded 98.7% of dye in 2 h. Calcination broadened the range of visible light responses, as well as conferred a high surface area, high Ag dispersibility, and low recombination rate of photogenerated e^−^/h^+^ recombination [128]. Porous graphitic carbon nitride (p-C_3_N_4_) has been prepared via a simple pyrolyzing treatment of g-C_3_N_4_, which introduces structural defects into the g-C_3_N_4_ via breaking some bonds, as shown in the Figure 6. The defects were found to be advantageous for the generation of e^−^/h^+^ charges and the prevention of the recombination of these charges. The p-C_3_N_4_ exhibited a narrow band gap for promoting visible light utilization as compared with g-C_3_N_4_. p-C_3_N_4_ degraded 90% of the MO dye in 90 min, which was sufficiently higher than the activity of g-C_3_N_4_ (only 19% after 90 min) under visible light [129]. Similarly, the results discussed in the doping section about the P and S-co-doped g-C_3_N_4_ indicate that P and S co-doping causes defects in the sample structure. The structure defects could trap photo-induced electrons, which can promote the e^−^/h^+^ separation, inhibiting the recombination of the photogenerated charges and prolonging their lifetime [121].

Similarly g-C_3_N_4_ molecules were also used for enhancing the photocatalytic activity of other materials, such as graphene aerogels [130], for the efficient photodegradation of MO dye.

## 5. Conclusions and Future Perspectives

The photodegradation of MO is the most effective method for its complete mineralization. g-C_3_N_4_ based materials are widely utilized in photo-catalytic materials owing to their low raw material prices, high physicochemical stability, easy and simple synthesis, earth-abundant nature and fascinating electronic band structure. The lower degradation efficiency of g-C_3_N_4_ can be enhanced via coupling with other materials to form composites and heterojunctions. Efficiency can also be enhanced through introducing structural defects, crystal phase control and doping with metals and non-metals. Preparation of composites and heterojunctions is the most followed approach, because the resulted materials exhibit improved photocatalytic efficiency, owing to their synergistic effects. Similarly, the structural defects were found to be beneficial for the e^−^/h^+^ generation and inhibition of the charge recombination.

There are a few dimensions that still need thorough investigations. Some of them are the following.

The majority of the g-C_3_N_4_ based photocatalysts are not economical and their preparation methods are quite complex. Therefore, exploring the unprecedented g-C_3_N_4-_based photocatalysts, which are economical, easy to prepare and have superior photocatalytic characteristics to cope with the industrial needs, is very important.

Each individual process of photodegradation of MO dye through g-C_3_N_4_ based photocatalysts needs to be visualized specifically through in situ characterization techniques. Through in situ characterizations, one can fabricate highly robust, cost-effective and perfectly photon energy-matched catalysts with commercial feasibility and effective photostability.

Monitoring of the photodegradation of dyes in aqueous solutions by recording the changes in UV-Vis absorption may not allow the accurate detection of the full degradation of organic pollutants. Therefore, it is suggested to use other monitoring techniques.

Mechanistic understanding of the PD of MO by g-C_3_N_4_-based photocatalysts has been examined, but researchers need to further investigate this with theoretical and computational and support. Through computational approaches, researchers are able to design g-C_3_N_4_-based photocatalysts for the efficient adsorption and photodegradation of MO dyes. Through this approach, researchers can also choose effective doping and coupling materials for g-C_3_N_4_, for the efficient adsorption and photodegradation of MO dyes. DFT calculations can suggest dye degradation mechanisms and are helpful for supporting the obtained experimental results.

## Figures and Tables

**Figure 1 molecules-28-03199-f001:**
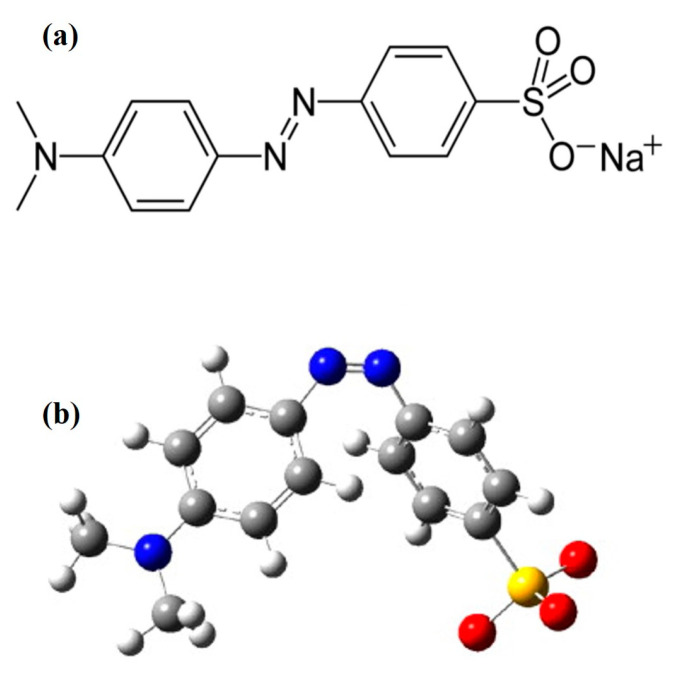
(**a**) Chemical structure of Methyl orange; (**b**) the optimized MO molecular structure created using Gaussian 03 software package on the basis of the HF/6-31G method. Reprinted/adapted with permission from [19], 2023, Elsevier (License Number 5507170462000).

**Figure 2 molecules-28-03199-f002:**
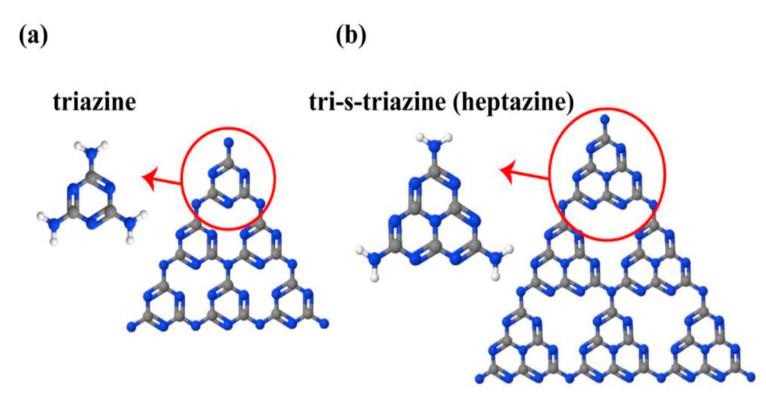
(**a**) Triazine and (**b**) tri-striazine (heptazine) structures of g-C_3_N_4_ (blue, gray and white balls are nitrogen, carbon and hydrogen, respectively) [60].

**Figure 3 molecules-28-03199-f003:**
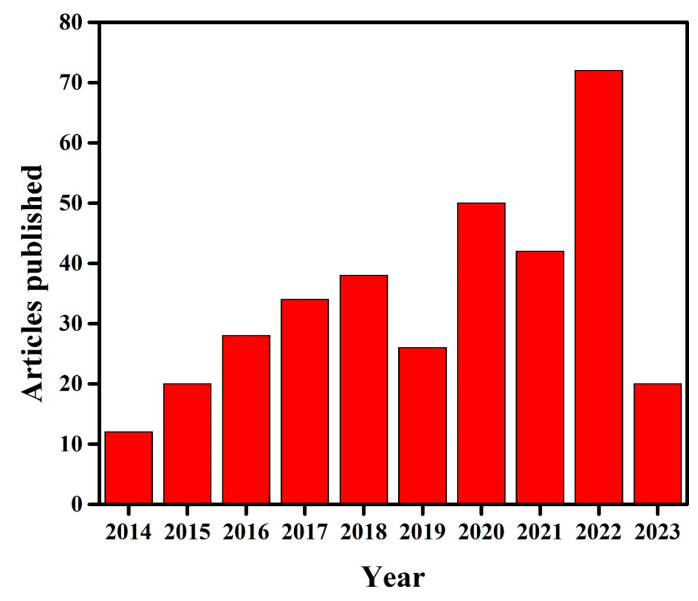
Annual article frequency published extracted from the Scopus database on 13 March 2023 (Searched with the keyword ‘g-C_3_N_4_ for methyl orange degradation’).

**Figure 4 molecules-28-03199-f004:**
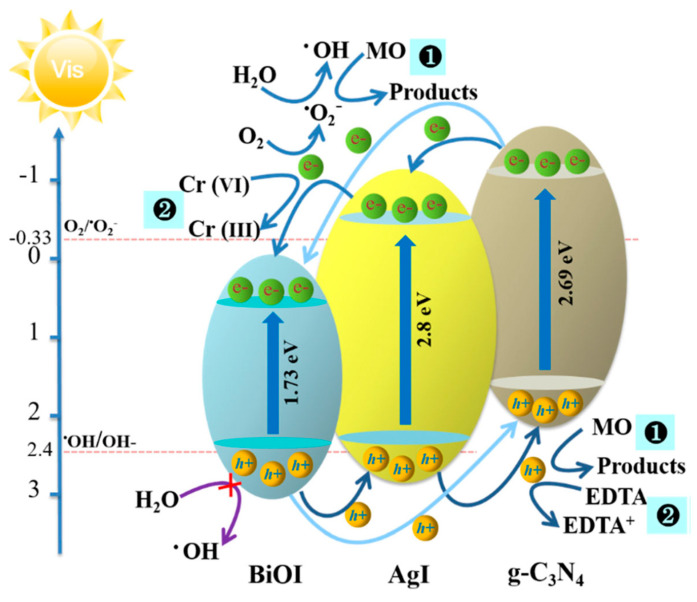
The possible proposed photocatalytic mechanism for MO dye degradation (path 1), showing the BiOI/AgI/g−C_3_N_4_ composites’ irradiation under visible−light. Reprinted/adapted with permission from [115], 2023, Springer Nature (License Number 5507180161444).

**Figure 5 molecules-28-03199-f005:**
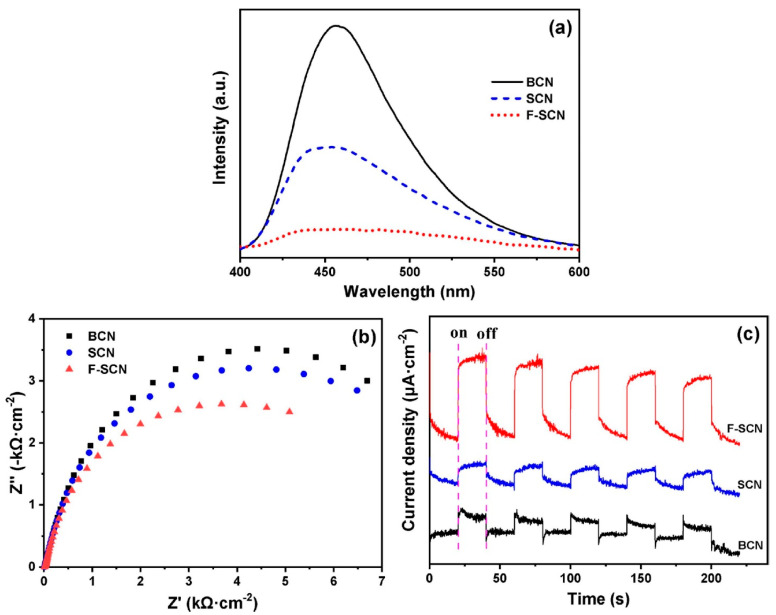
(**a**) PL spectra, (**b**) EIS Nyquist spectra, and (**c**) transient photocurrent response of BCN, SCN, and F-SCN. Reprinted/adapted with permission from [118], 2023, Elsevier (License Number 5507180566714).

**Figure 6 molecules-28-03199-f006:**
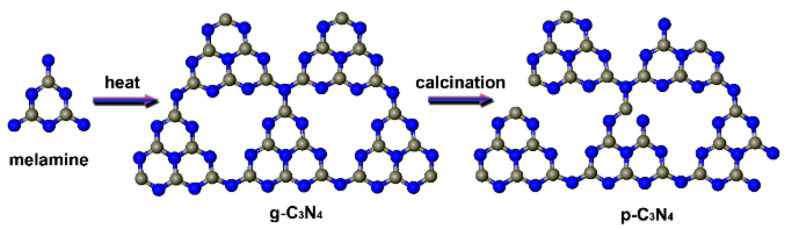
Schematic diagram for the formation of p-C_3_N_4_ [129].

**Table 1 molecules-28-03199-t001:** g-C_3_N_4_ based composites/nanocomposites used for the PD of MO.

Composites with Preparation Method	Efficient Composite	%Degradation of MO	Reason	Ref.
Z-scheme HOFs/g-C_3_N_4_ heterojunction. In situ electrostatic method	PFC-1/CNNS heterojunction	100%/60 min under the visible light irradiation	The heterojucntion inhibits photo-generated e^—^h^+^ recombination to produce more highly reactive species (^•^OH, ^•^O_2_^−^ and h^+^)	[93]
g-C_3_N_4_/Bi_2_WO_6_ composites. Hydrothermal method	g-C_3_N_4_:Bi_2_WO_6_ (0.5:1) hydrothermal time 6 h	94.82%/ 180 min.	Formation of g-C_3_N_4_–Bi_2_WO_6_ heterojunctions between Bi_2_WO_6_ nanoparticles and g-C_3_N_4_ sheets.	[94]
Ag_3_PO_4_/Ti-BDC/g-C_3_N_4_ ternary composites. Mixing of the constituents components.	Ag_3_PO_4_/Ti-BDC/g-C_3_N_4_-3% ternary composite	99%/60 min	Multi-structural and synergistic effects. Some Ag^0^ was generated, which contributes to slowing the e^−^/h^+^ recombination	[95]
PES-Ag_3_PO_4_/g-C_3_N_4_ Film. Non-solvent-induced phase inversion process	PES-Ag_3_PO_4_/g-C_3_N_4_ (13%)	97%/180 min	g-C_3_N_4_ improved the photo-catalytic efficiency of Ag_3_PO_4_ and increased hydrophilicity of photo-catalyst film.	[96]
BaTiO_3_@g-C_3_N_4_ composites. Simple mixing–calcining method	BaTiO_3_@g-C_3_N_4_(12%)	76%/6 h	Efficient separation of the photogenerated e^−^/h^+^ pairs due to the carrier migration between BaTiO_3_ and g-C_3_N_4_.	[97]
Mn@PC/g-C_3_N_4_	20% Mn@PC/g-C_3_N_4_	---	Improved light response and light-harvesting ability of the Mn@PC/g-C_3_N_4_ and effective charges separation.	[98]
g-C_3_N_4_/ZnS composite. Hydrothermal process	10 wt% g-C_3_N_4_/ZnS nanocomposite	93.0 %/100 min	The g-C_3_N_4_ nanosheets serve as electron mediators as well as transporters to lengthen the lifetime of created charges in the composite. Moreover, the e^—^h^+^ pair recombination of ZnS has been efficiently restricted.	[99]
ZnO/Ph-g-C_3_N_4_ nanocomposite. Single-step calcination and combustion process	0.05% ZnO/Ph-g-C_3_N_4_	97.7%/120 min	Heterojunction results in improved absorption of light and efficient e^−^/h^+^ pair separation.	[100]
(CNS-TiO_2_/g-C_3_N_4_) photocatalyst. One step hydrothermal and calcination methods.	CNS-TiO_2_/gC_3_N_4_-2	99.8%/80 min	Z-Scheme heterojunctions in CNS-TiO_2_/g-C_3_N_4_ promote the photo-generated e^−^/h^+^ ratio as well as catalyst lifetime.	[101]
g-C_3_N_4_/Bi_2_WO_6_ composite photocatalysts.	70 wt% gC_3_N_4_/Bi_2_WO_6_.	100%/180 min	Synergic effect between Bi_2_WO_6_ and g-C_3_N_4_ improved photo-induced carrier separation.	[102]
NG@g-C_3_N_4_ nanocomposite. Ultra-sonication method.	NG@g-C_3_N_4_ nanocomposite	100%/35 min	High visible light adsorption capability, improved charge separations and high surface area owing to the synergistic effect of the NG grafted to g-C_3_N_4_.	[103]
CdS-WPB–g-C_3_N_4_ heterojunction. simple hydrothermal synthesis	CdS-WPB(5%)-CN composite	99%/80 min under visible light	WPB effectively transfer photogenerated electrons at the interface between g-C_3_N_4_ and CdS.	[104]

## Data Availability

Not applicable.

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
