# Peer review of "g-C3N4 Based Photocatalyst for the Efficient Photodegradation of Toxic Methyl Orange Dye: Recent Modifications and Future Perspectives"

_molecules, 2023, doi:10.3390/molecules28073199_

Round 1

Reviewer 1 Report

The authors should pay attention to the issues as follows.

1 The authors should check through the whole article and revise to make the statements more logical. For example, I think the statements in line 99-101 should be placed in the passage above.

2 The formats in line 123-124, 233-239 are wrong.

3 The authors should check through the whole article and revise the mistakes. For example, nanocompost in line 140.

4 The formats of literatures should be kept consistent. The first letters in some titles are  upper case, while some sre lower case. 

Author Response

Reviewer 1

Comments and Suggestions for Authors

The authors should pay attention to the issues as follows.

Author`s Response: Thank you for reviewing our review article in a critical way and your valuable comments. Incorporation of the suggested changes will further improve the scientific quality of our article. Your comments were addressed on a point-by-point basis.

1 The authors should check through the whole article and revise to make the statements more logical. For example, I think the statements in line 99-101 should be placed in the passage above.

Author`s Response: Thanks for your valuable comments. The mentioned line is placed in the above passage.

2 The formats in line 123-124, 233-239 are wrong.

Author`s Response: The mentioned text were formatted according to the journal text style.

3 The authors should check through the whole article and revise the mistakes. For example, nanocompost in line 140.

Author`s Response: The whole review article is thoroughly checked for correcting such mistakes.

4 The formats of literature should be kept consistent. The first letters in some titles are upper case, while some are lower case.

Author`s Response: Thanks for such nice comments. The formats of literature were thoroughly checked for consistency.

Reviewer 2 Report

Abdulelah Aljuaid et al. reported “g-C3N4 based Photocatalyst for the Efficient Photodegradation of toxic Methyl Orange Dye: Recent modifications and future perspectives”. The review article is publishable after addressing the following issues.

1.       Avoid using keywords that are the same as the titles words as much as possible.

2.       What are the maximum permissible limits of dyes? They should be discussed and compared to the concentrations being actually discharged.

3.       The lifecycle of dye in different ecosystems and the bio-magnification of these pollutant should be discussed.

4.       The governmental rules, regulations, and policies imposed across the world to control the release of these toxic compounds should be discussed.

5.       Why you select only one dye MB? Such type of catalyst is used for more such dyes. should be included such type of photo-catalyst for other dyes too.

6.       The negative effects of dyes on the environment, organisms, and human health should be further elaborated (eg: negative impacts on the endocrine and reproductive systems in both aquatic organisms and humans) in a more organized way in the introduction section.

7.       The regeneration or recycling studies must be covered in this review to demonstrate the efficiency of the investigated catalysts after being reused for several times.

8.       The economic factor should be discussed in this review by clarifying the production cost of the investigated materials.

9.       Clarify their advantages and disadvantages.

10.   Read and cite the following papers.

Molecules 2023, 28, 1081. https://doi.org/10.3390/molecules28031081

https://doi.org/10.1016/j.jece.2021.105580

https://doi.org/10.1016/j.jece.2019.103291

11.   The English language must be improved. 

Author Response

Reviewer 2

Comments and Suggestions for Authors

Abdulelah Aljuaid et al. reported “g-C3N4 based Photocatalyst for the Efficient Photodegradation of toxic Methyl Orange Dye: Recent modifications and future perspectives”. The review article is publishable after addressing the following issues.

Author`s Response: Thank you for reviewing our review article in a critical way and your valuable comments. Incorporation of the suggested changes will further improve the scientific quality of our article. Your comments were addressed on point-by-point basis.

  1. Avoid using keywords that are the same as the titles words as much as possible.

Author`s Response: The keywords are changed according to your suggestion.

  1. What are the maximum permissible limits of dyes? They should be discussed and compared to the concentrations being actually discharged.

Author`s Response: The permissible level of some dyes are mentioned in the introduction section as well as compared with the concentration actually discharged. The concentration of MO dye reported in textile effluent are also mentioned as 500 ppm.

  1. The lifecycle of dye in different ecosystems and the bio-magnification of these pollutant should be discussed.

Author`s Response: We have mentioned the stability of dyes in aqueous environment and their resistance to biodegradation.

  1. The governmental rules, regulations, and policies imposed across the world to control the release of these toxic compounds should be discussed.

Author`s Response: Dye removal is a big challenge overall the world due to industrialization and every country and government are trying to minimize as it is possible. We have tried to answer your this question in the review article.

  1. Why you select only one dye MB? Such type of catalyst is used for more such dyes. should be included such type of photo-catalyst for other dyes too.

Author`s Response: We have select MO (methyl orange) not MB (methylene blue). Yes this type of (g-C3N4) is used for the photodegradation of more other dyes, but we have specified it only to MO dye because reviewing this catalyst for other dyes cannot be consolidated in a single review.

  1. The negative effects of dyes on the environment, organisms, and human health should be further elaborated (eg: negative impacts on the endocrine and reproductive systems in both aquatic organisms and humans) in a more organized way in the introduction section.

Author`s Response: The negative effects of dyes on the environment, organisms, and human health are further elaborated following your valuable suggestion

  1. The regeneration or recycling studies must be covered in this review to demonstrate the efficiency of the investigated catalysts after being reused for several times.

Author`s Response: The regeneration/recycled for some catalysts are covered as suggested.

  1. The economic factor should be discussed in this review by clarifying the production cost of the investigated materials.

Author`s Response: The use of g-C3N4, as an economical point of view is discussed in detail. Their economical raw materials for their synthesis are also mentioned.

  1. Clarify their advantages and disadvantages.

Author`s Response: The advantages and disadvantages of g-C3N4 as photocatalysts are already discussed and further elaborated.

  1. Read and cite the following papers.

Molecules 2023, 28, 1081. https://doi.org/10.3390/molecules28031081

https://doi.org/10.1016/j.jece.2021.105580

https://doi.org/10.1016/j.jece.2019.103291

Author`s Response: All these articles were cited which further improved the scientific quality of our review article.

  1. The English language must be improved.

Author`s Response: The language has been improved

Reviewer 3 Report

The work of Aljuaid et al. is a review on the photodegradation of methyl orange (MO) using g-C3N4-based photocatalysts, which is in the beginning complemented by reviewing different composites or heterojunctions devoted to the preparations, functions, and mechanisms. The authors have summarized the process of realizing g-C3N4-based photocatalysts for the MO photodegradation.

Before publication, the authors are suggested to consider the following issues:

(1) The authors should give their perspectives on the formation of g-C3N4-based heterojunctions by reviewing the corresponding literature, which benefits the understanding of the improved photocatalytic activity. For example, the formation of special interfaces via chemical bonds, etc. Please refer to the ref. 10.1002/sstr.202100068

(2) In Table 1, the authors have listed a summary of g-C3N4-based photocatalysts for MO. It will be better to discuss the advantages and disadvantages of these nanocomposites, including the preparation techniques, photocatalytic activities, etc.

(3) Most of these publications focus on the photodegradation of dyes in aqueous solutions by recording the changes in UV-Vis absorption, actually, which does not allow accurate detection of the full degradation of organic pollutants (10.3390/nano12071074). Do the authors have any comments on this phenomenon from the point of the photodegradation of MO using g-C3N4-based photocatalysts?

(4) I suggest the authors merge “Future perspectives” and “Conclusion” as “Conclusion and Future perspectives”, in which the authors should not only summarize this paper but also point out the current challenges, as well as some possible suggestions for future research.

(5) Extensive editing of the English language and style is required.

Author Response

Reviewer 3

Comments and Suggestions for Authors

The work of Aljuaid et al. is a review on the photodegradation of methyl orange (MO) using g-C3N4-based photocatalysts, which is in the beginning complemented by reviewing different composites or heterojunctions devoted to the preparations, functions, and mechanisms. The authors have summarized the process of realizing g-C3N4-based photocatalysts for the MO photodegradation.

Before publication, the authors are suggested to consider the following issues:

Author`s Response: Thank you for reviewing our review article in a critical way and your valuable comments. Incorporation of the suggested changes will further improve the scientific quality of our article. Your comments were addressed on point-by-point basis.

(1) The authors should give their perspectives on the formation of g-C3N4-based heterojunctions by reviewing the corresponding literature, which benefits the understanding of the improved photocatalytic activity. For example, the formation of special interfaces via chemical bonds, etc. Please refer to the ref. 10.1002/sstr.202100068

Author`s Response: Thanks for your comments. The on the formation of g-C3N4-based heterojunctions and their advantages and benefits are already discussed in the section 2 and Table 1.

(2) In Table 1, the authors have listed a summary of g-C3N4-based photocatalysts for MO. It will be better to discuss the advantages and disadvantages of these nanocomposites, including the preparation techniques, photocatalytic activities, etc.

Author`s Response: The preparation tecniques, photocatalytic activities and the advantages (in reason column) already listed in the Table, however in these studies their disadvantages are not mentioned.

(3) Most of these publications focus on the photodegradation of dyes in aqueous solutions by recording the changes in UV-Vis absorption, actually, which does not allow accurate detection of the full degradation of organic pollutants (10.3390/nano12071074). Do the authors have any comments on this phenomenon from the point of the photodegradation of MO using g-C3N4-based photocatalysts?

Author`s Response: In most of the literature cited in our review article focuses on monitoring photodegradation of MO dye by using UV-Vis adsorption. However I have included this point in our article future perspectives.

(4) I suggest the authors merge “Future perspectives” and “Conclusion” as “Conclusion and Future perspectives”, in which the authors should not only summarize this paper but also point out the current challenges, as well as some possible suggestions for future research.

Author`s Response: Following your valuable suggestion, both sections were merged.

(5) Extensive editing of the English language and style is required.

Author`s Response: The English has been sufficiently improved.

Round 2

Reviewer 2 Report

The author addressed all the issues in the revised manuscript and a I am going to recommend the manuscript for publication in the present form.

Reviewer 3 Report

It can be accepted in its present form now.